# The Effectiveness of Different Cleaning Methods for Clear Orthodontic Aligners: Impacts on Physical, Mechanical, and Chemical Properties—An In Vivo Study

**DOI:** 10.3390/polym17121620

**Published:** 2025-06-11

**Authors:** Athar Alweneen, Nasser Alqahtani

**Affiliations:** Department of Pediatric Dentistry and Orthodontics, College of Dentistry, King Saud University, Riyadh 11545, Saudi Arabia; nasserdm@ksu.edu.sa

**Keywords:** aligners, orthodontics, cleaning method, color stability, thermoplastic material

## Abstract

Maintaining the cleanliness of orthodontic aligners is crucial for oral hygiene and preserving the optical properties of aligners. In this randomized clinical trial, we compared the effectiveness of different cleaning methods for the maintenance of Invisalign clear aligners. Twelve adult patients received five aligners, each worn for 10 days. The aligners were divided based on the cleaning method: tooth brushing with whitening toothpaste, vinegar, Fittydent Super Cleansing Tablets, Invisalign cleaning crystals, and only water. Scanning electron microscopy (SEM) was used to detect surface morphology changes; color changes (ΔE) were evaluated using a spectrophotometer. Fourier transform infrared spectroscopy (FTIR) with a diamond hemisphere was used to study the aligners’ chemical compositions. Nanoindentation testing was used to assess changes in the elastic modulus. SEM confirmed the effectiveness of Invisalign cleaning crystals in maintaining cleanliness, revealing a surface similar to that of the control group with no adverse effects. Color stability analysis revealed significant ΔE value differences; whitening toothpaste had significantly lower ΔE values than water and Invisalign cleaning crystals. The elastic modulus and FTIR analyses indicated no significant differences between the cleaning methods. Therefore, Invisalign cleaning crystals and whitening toothpaste are safe for aligner maintenance, showing successful and aesthetically pleasing results.

## 1. Introduction

A clear aligner is an orthodontic treatment modality that is both esthetically pleasing and effective, meeting the needs of patients [1]. Each aligner typically remains in the oral cavity for 1–2 weeks before being replaced by a new aligner, facilitating gradual tooth movement. The stability of the aligner’s properties during this short timeframe is essential for clinical efficacy [2,3]. These removable appliances are desired by adults seeking a more esthetically pleasing option with less metal exposure [4]. The fabrication of a sequential removable aligner involves the use of a polyurethane thermoplastic material. This process utilizes high-precision stereolithography and milling technologies [2], and the generated series of aligners have the ability to induce tooth movement [5]. Clear aligners have some drawbacks when compared to fixed orthodontic appliances. Unlike metal and ceramic braces, which apply force continuously, clear aligners rely on patient compliance to be effective, needing 20–22 h of use each day. Patients are typically instructed to continuously wear their aligners, with the exception of during mealtimes, when consuming anything other than water, and while brushing or flossing their teeth [6].

Despite the promising nature of clear aligners as esthetically pleasing orthodontic tools, information regarding the esthetic stability of their constituent materials is lacking. Clear aligners may be more susceptible to staining, distortion, and decreased mechanical properties as a result of dietary habits and cleaning procedures, compromising their transparency and performance over time. Metal and ceramic braces, on the other hand, do not need to be replaced as frequently and are less likely to discolor or distort [7]. From an esthetic perspective, the color stability and transparency of clear orthodontic aligners should remain consistent throughout the 2-week orthodontic treatment period. However, the color durability of dental materials is frequently affected by various factors, including exposure to ultraviolet light, staining with drinks, and the use of mouthwashes [8]. Aligners can also affect the oral environment and increase the risk of bacterial accumulation and related consequences, such periodontal disease, halitosis, and carious lesions [9]. Unlike traditional braces, which are cleaned during dentist visits, removable orthodontic aligners should be cleaned before insertion to prevent bacterial accumulation on their surfaces [9]. Furthermore, many patients do not follow the orthodontists’ and manufacturers’ instructions and consume colored beverages containing chemicals while wearing aligners [4], which leads to changes in the polymer present in the aligners, thereby compromising one of their primary advantages, transparency [10]. Accumulation of pigments in the staining agents may alter the color of the aligner materials. Hence, transparent aligners may lose some of their visual appeal during the 2-week treatments [7]. A key problem with aligners is that their internal surfaces are susceptible to plaque accumulation and staining, owing to the stagnation of salivary flow [11]. Although calculus formation is uncommon, it is not impossible [12]. Therefore, maintenance of excellent oral hygiene while wearing aligners is considered crucial [3]. Proper cleaning of orthodontic aligners is crucial not only to prevent bacteria from entering the oral cavity but also to ensure that these appliances, which are generally selected to meet esthetic objectives and are thus expected to be visually appealing, remain bright and odorless over time [13].

With the advent of removable orthodontic devices, patients can practice greater control over their dental hygiene and access a standard and professional level of hygiene-related cleaning, thereby reducing the possibility of acquiring diseases linked to plaque accumulation [14]. The available cleaning products can effectively remove bacterial biofilms, restore the appliance’s original transparency, and provide a pleasant odor when immersed in water for a few minutes daily [15]. However, the reactivity of cleansers has prompted concerns regarding potential chemical modifications to the aligners and, subsequently, their mechanical properties, which could impede the positive outcome of the treatment. Thus, the application of appropriate cleaning techniques is crucial for preserving the structural integrity of the thermoplastic materials comprising aligners.

Nevertheless, the information available in the literature regarding the adverse effects of this interaction is limited. While several studies have looked into the impact of cleaning procedures on thermoplastic retainers used to stabilize the orthodontic treatment outcome [10,15,16], few studies have compared different cleaning techniques for Invisalign clear aligners (Align Technology, Santa Clara, CA, USA) under standardized settings. Five cleaning methods commonly available to patients for Invisalign clear aligners include tooth brushing with whitening toothpaste, 2.5% acetic acid (vinegar), Fittydent Super Cleansing Tablets (Fittydent International Inc., Wien, Austria), Invisalign^®^ cleaning crystals (Align Technology Inc., San Jose, CA, USA), and water. Despite prior research on stain formation and cleaning techniques, no comprehensive study has compared these five methods under standardized conditions. As not all patients wearing aligners are diligent in terms of avoiding coloring agents and cleaning their aligners properly, this study aimed to assess the changes in the physical, mechanical, and chemical properties of Invisalign clear aligners using these five different cleaning methods. Moreover, determining the superior method for cleaning aligners without changing their physical, mechanical, and optical characteristics is vital. The hypothesis tested in this study was that there is no significant difference in the effectiveness of the five cleaning methods when applied to Invisalign clear aligners.

## 2. Materials and Methods

This study was approved by the institutional review board of King Saud University College of Medicine (E-23-7820). All the participants provided written informed consent for participation in the study and publishing their data.

In this randomized crossover trial, all the participants tested each cleaning method using separate aligners in a randomized order. The study design allowed for comparisons among participants, potentially reducing bias. Power calculations were performed, and for an effect size of 0.71, a power of 0.85, and an alpha value of 0.05, the sample size was found to be 60. As a result, 12 patients were selected and given 5 aligners, each corresponding to a specific cleaning method. Patients were selected from a list of those who had already begun their Invisalign clear alignment treatment at the Dental University Hospital, King Saud University, Riyadh, Saudi Arabia. The patients underwent a full-mouth examination, which included a thorough dental and medical history. Patients who satisfied the inclusion criteria, including those aged ≥18 years, with good oral hygiene, no systemic diseases or active periodontal disease, and no active caries, were provided with detailed information about participation in this study. Patients with a history of smoking and debilitating health conditions interfering with patient compliance were excluded.

Prior to starting the study, the principal investigator reviewed the informed consent forms of patients who had the opportunity to ask questions. The principal investigator also motivated and instructed all the patients regarding proper oral hygiene. The patients were instructed to use a manual toothbrush with a rolling technique and were provided with the same oral hygiene products to reduce bias (toothpaste and interdental floss). Each patient received a series of five aligners; each aligner was to be worn for 10 days to accommodate patient compliance. Patients were also provided with the materials required for the different cleaning methods in labeled plastic bags. Each plastic bag contained instructions on the front and tracking on the back for monitoring. The tracking checklists were marked daily, and once all the checks were completed, the patients were informed of the principal investigator. The materials were divided into five groups according to the cleaning method used over the course of approximately 7 weeks, where the materials for each method were returned to the labeled plastic bags after use. Patients were instructed to use the following cleaning methods once every day:Group A (tooth brushing with whitening toothpaste): during the first 10 days, each patient was instructed to remove their aligner and brush it for a minimum of 30 s with a soft toothbrush and a whitening toothpaste containing hydrogen peroxide and a dentin abrasion value of less than 100.Group B (2.5% acetic acid and vinegar): during the second 10 days, each patient was instructed to soak their aligner in a 100 mL vinegar solution diluted in 100 mL of distilled water for 15 min [15].Group C (Fittydent Super Cleansing Tablets): During the third 10 days, Fittydent Super Cleansing Tablets containing sodium bicarbonate (Fittydent International, Inc., Wien, Austria) were used. Each patient was instructed to remove the aligner and rinse it using tap water. A clean container filled with 100 mL of warm water and one tablet were used, with the aligner completely soaked and left for 10 min, followed by rinsing and drying.Group D (Invisalign cleaning crystals): During the fourth 10 days, Invisalign cleaning crystals (Align Technology Inc., San Jose, CA, USA) were used containing sodium bicarbonate and sodium sulfate. Each patient was instructed to remove their aligner and rinse it using tap water. A clean container filled with 100 mL of warm water and one packet of cleaning crystals was used, with the aligner fully immersed in the cleaning crystal solution. The patient was then instructed to shake the container gently for 20 s to ensure the distribution and dissolution of the crystals, and the aligner was left for 15 min, followed by rinsing and drying.Group E (water only): during the last 10 days, each patient was instructed to remove their aligner and brush it without toothpaste.

At the end of the study, multiple analyses were performed using a brand-new as-received aligner with no history of intraoral exposure as a reference (control) to compare the effects of intraoral exposure and different cleaning methods on the tested aligners.

### 2.1. Labeling System

As mentioned above, each patient was provided with plastic bags clearly labeled with the corresponding cleaning methods in both English and Arabic and instructed to return each aligner that had been cleaned with a specific cleaning method in the plastic bag labeled with the relevant cleaning method.

### 2.2. Scanning Electron Microscopy

For the scanning electron microscopy (SEM) analysis, the aligner samples from the different groups were fixed to stubs with double-sided carbon tape. A thin layer of gold was then sprayed onto the samples using a quorum sputter coater (Q150RS, Laughton, East Sussex, UK) under vacuum and argon atmosphere for 1 min (Figure 1A). The surface morphology of each aligner sample was examined at an accelerating voltage of 15 kV by using a Zeiss device (EVO-LS10, Cambridge, UK). Images were captured using the Smart SEM software, version 5.05 (Zeiss, Cambridge, UK) (Figure 1B).

The surface roughness of each captured image (at 10,000× magnification) was qualitatively evaluated using the ImageJ software (version 1.53 t, NIH, Bethesda, MD, USA). The desired image was opened in the ImageJ software, and the scale was set according to the micron marker. The complete SEM image was selected and analyzed using the surface plot plugin of the ImageJ software. The resulting three-dimensional (3D) plots were saved and grouped with the corresponding SEM images.

### 2.3. Spectrophotometry

The color coordinates of each sample were measured using a spectrophotometer (LabScan-XE^®^, Reston, VA, USA). Each prepared sample was positioned over the spectrophotometer sensor, and three observations were performed for each sample. The instrument transmitted light waves to the test specimens, and the refracted or reflected light waves were quantified using the L*a*b* color coordinates. The “L” value for each scale indicates the level of light/darkness, while the “a” and “b” values indicate the redness/greenness and yellowness/blueness, respectively. All the three values were obtained to facilitate a comprehensive description of the hue of each specimen. The color stability was determined by calculating the delta E (ΔE) from the Commission Internationale de I’Eclairage (CIE) L*a*b* values using the following formula:ΔE = [(ΔL*)2 + (Δa*)2 + (Δb*)2]1/2(1)

To provide a clinical interpretation, the data were converted to the National Bureau of Standards (NBS) system (Table 1) using the following equation [17,18]:NBS = ∆E* × 0.92.(2)

### 2.4. Nanoindentation Test

Nanoindentation testing of the surface of each sample was conducted using a nanomechanical device indenter (UMT1, Bruker, San Jose, CA, USA) equipped with a Berkovich diamond indenter, with a maximum load of 10 mN. Each sample underwent four indentations, and the mean value was calculated for each sample. The elastic modulus was determined using the CETR UMT (universal mechanical tester) software (version 2.2.107), including a nanoindentation apparatus.

### 2.5. Fourier Transform Infrared Spectroscopy

The Fourier transform infrared (FTIR) spectra of the different patient groups, each treated with a different cleaning method, were analyzed and recorded using a Bruker Alpha II FTIR-attenuated total reflectance (ATR) spectrophotometer (Bruker, Bremen, Germany) equipped with a platinum diamond crystal for the ATR analysis. Data processing was performed using the advanced Bruker Optics Opus 7.8 software (Bruker Optik GmbH, Ettlingen, Germany). A uniform portion of each sample was placed in the measuring channel of the diamond sphere, allowing the IR beams to pass through the diamond and reflect from the sample surface. The FTIR-ATR wave numbers were recorded in the range of 400–4000 cm^−1^, with a spectral resolution of 2 cm^−1^. The Bruker Opus software presented the data in percentage transmittance (%T) across various wave numbers, generating each FTIR-ATR spectrum.

### 2.6. Data Analysis

Data were analyzed using the SPSS version 29.0 statistical software (IBM Inc., Chicago, IL, USA). Descriptive statistics (mean and standard deviation) were used to describe the outcome variables (elastic modulus and ΔE). As the NBS values did not follow a normal distribution and the *p*-value of the Shapiro–Wilk test was *p* < 0.05, the data were reanalyzed using a non-parametric test (Kruskal–Wallis test) to compare the mean ranks among the study groups, followed by the Conover post hoc test for a pairwise comparison of each group with other groups in the study. Student’s *t* test was used for independent samples, while a one-way analysis of variance followed by a post hoc test (for multiple pairwise comparisons) was used to compare the mean values of two outcome variables between the study groups and between each study group and the control group. A *p*-value of ≤0.05 was used to report the statistical significance of the results.

## 3. Results

### 3.1. SEM

Following the various cleaning procedures, the aligner samples were examined using a scanning electron microscope to evaluate the material integrity, microbial load, and surface cleanliness. The surface roughness parameters of the aligner samples from the different groups are presented in Table 2.

The control group exhibited a clean surface free of microbiological load, because the oral cavity was never exposed. The surface plot indicates that the sample surfaces of the control group were slightly rough, which could be attributed to the manufacturing process (Figure 2). Group A displayed the lowest values for Rq and Ra, indicating the smoothest surface among all groups with a clear surface and very little debris or plaque, suggesting that the aligners were effectively cleaned using a toothbrush, whitening toothpaste, and mild brushing (Figure 3). The Group B sample surfaces were clear of debris and plaques. The surface plot of the Group B samples also confirmed moderate roughness after cleaning with the 2.5% acetic acid/vinegar solution; the roughness values showed similar trends with the control but were slightly smoother (Figure 4). The SEM images of the Group C (cleaned using the Fittydent Super Cleansing Tablet solution) samples revealed a clear debris/plaque-free surface but increased surface roughness. The values of Group C were similar to those of Group B, but they were slightly smoother than those of the control group, as observed in the 3D surface plot of SEM images (Figure 5). The strong cleaning agents found in Fittydent Super Cleansing Tablets, such as citric acid, sodium bicarbonate, and peroxide, may degrade the thermoplastic material used to fabricate Invisalign aligners. The SEM images of the Group D samples (cleaned using the Invisalign cleaning crystal solution) clearly depicted no residual plaque or any visible bacterial colony at a very high magnification (10,000×). The Group D sample surfaces exhibited the highest Rq, indicating a rougher surface. The surface was further confirmed by the 3D surface plots of the Group D samples (Figure 6). The SEM images of the Group E samples, which were cleaned using only water and a toothbrush with no toothpaste, clearly revealed stain marks and deposits of food debris or plaque on the aligner surface (Figure 7). No bacterial or fungal colonies were detected. Water alone was unable to remove the stain marks. The 3D surface plot of the Group E samples demonstrated a rougher surface than that of the control group, confirming ineffective cleaning. A comparison of the treatment groups with the control group confirmed that the use of Invisalign cleaning crystals was the most effective cleaning method for aligners in terms of the presence of plaque and debris with the preservation of surface toughness as the control group. However, these results are only qualitative, indicating the need for an effective quantitative method for obtaining better insights.

### 3.2. Spectrophotometry

A comparison of the mean ΔE values among the five study groups demonstrated a highly statistically significant difference (F = 8.661, *p* < 0.0001). Multiple comparisons of the mean values in each pair of study groups demonstrated statistically significant differences between the mean ΔE values, as well as between Group B and three other study groups. Notably, the mean ΔE values of Group B were significantly higher than those of Groups A, C, and D (*p* < 0.001) but were not different from the mean values of Group E (*p* = 0.139).

Moreover, no significant difference was observed in the mean ΔE values among Groups C, D, and E (*p* = 0.780), while the mean ΔE values of Group E were significantly higher than the mean values of Group A (*p* < 0.001). In addition, the mean ΔE values of Group D were significantly higher than the mean values of Group A (*p* < 0.001). In contrast, no statistically significant difference in the mean ΔE values was noted between Groups A and C (*p* = 0.095) (Table 3).

### 3.3. NBS Rating

To correlate the amount of color change (ΔE) recorded by the spectrophotometer to the clinical environment, the data were converted into NBS units. The results of this conversion are shown in Table 4 and described here in terms of NBS-defined color differences. According to the NBS ratings, after the aligners had been used and cleaned, all five cleaning methods were associated with marked color changes. However, brushing with whitening toothpaste resulted in perceivable color changes.

### 3.4. Nanoindentation

A comparison of the mean elastic modulus values among the five study groups demonstrated no statistically significant difference (F = 0.512, *p* = 0.727). Thus, no evidence indicated that the mean elastic modulus values were significantly different among the five study groups. Additionally, no statistically significant differences were observed between the mean elastic modulus values of the five study groups and that of the control group. Table 5 shows the mean difference in the elastic modulus between each of the five study groups and the control group and the associated *p*-values. All *p*-values were >0.05, indicating no statistically significant differences in the mean values.

### 3.5. FTIR

We conducted comparative studies on the aligners before and after the cleaning treatment using the spectrum obtained before cleaning as the reference standard. The FTIR data of the 12 aligners treated with the 5 different cleaning methods (Groups A–E) were subjected to FTIR-ATR analysis. This analysis revealed characteristic peaks, including N–H stretching of the secondary amines at 3310 cm^−1^ and a band at 1600 cm^−1^ due to N–H bending deformation vibrations. Additionally, the peak at 1698 cm^−1^ indicated the presence of C=O, while the peak at 1411 cm^−1^ was associated with CH_2_. The peak at 1216 cm^−1^ corresponded to strong C–O stretching in the esters. All samples across the five cleaning methods (Groups A–E) exhibited the same characteristic peaks. No changes were detected in any of the samples analyzed. Based on the representative examples of the groups, a comparison of the standard with the data in Figure 8 reveals that across the five groups, the spectra showed similar peaks, corresponding to the standard peaks at various wave numbers between 4000 cm^−1^ and 400 cm^−1^. Although all the aligners exhibited the same characteristic peaks before and after cleaning, the intensities varied. For example, the C=O peak at approximately 1700 cm^−1^ demonstrated an intensity of approximately 40% transmittance in the Figure 8 sample before treatment. After cleaning, the C=O peak intensities ranged from 48 to 65%. This reduction in intensity was likely attributed to degradation following the treatment. Similar variations in intensity were observed for other peaks across all the analyzed samples in the FTIR spectra, which were attributed to degradation after treatment.

The variation in transmittance among differently treated aligners can be attributed to the nature of the substances used. Group B treated with 2.5% acetic acid (vinegar)—a polar compound—shows reduced transmittance due to the influence of hydrogen bonding. Group A and C, treated with brushing with whitening toothpaste and Fittydent Super Cleansing Tablets, respectively, contain similar components, such as sodium bicarbonate and sodium lauryl sulfate, which contribute to surface erosion and result in comparable transmittance levels. In contrast, aligners treated with water and Invisalign cleaning crystals exhibit the highest transmittance, likely due to minimal interference with the molecular structure of the aligner material.

## 4. Discussion

The use of clear aligners in orthodontic treatment is common, because they are an esthetically pleasing and nearly invisible treatment option, and patients treated with removable aligners demonstrate a high level of compliance with oral hygiene practices, which decreases the risk of plaque-related diseases [19]. Nevertheless, it is essential to provide patients with a comprehensive at-home hygiene routine for intraoral appliances, particularly orthodontic aligners. The findings of the present study, as demonstrated by SEM analysis, confirmed the effectiveness of Invisalign cleaning crystals in maintaining the cleanliness of aligners, revealing a similar surface with the control with no adverse effects. Color stability analysis revealed significant differences in the ΔE values, with vinegar causing the most discoloration, while whitening toothpaste resulted in significantly lower ΔE values than water and Invisalign cleaning crystals. The cleaning methods based on Fittydent Super Cleansing Tablets, Invisalign cleaning crystals, and water demonstrated no significant differences in the mean ΔE values. The elastic modulus and FTIR analyses indicated no significant differences between the cleaning methods. The mechanical characteristics of the aligners must remain consistent during the in-service period for predictable tooth movement. However, the findings of the current study demonstrated that some cleaners may affect the mechanical characteristics and surface chemistry of the aligner materials.

According to a study performed for assessing and recording short-term chemical and physical changes during use, Invisalign appliances undergo morphological and structural modifications. Owing to poor alignment hygiene, appliances may become pigmented, losing their gloss and transparency. After 14 days of use, the aligners displayed microcracks, abraded and delaminated areas, localized calcified biofilm deposits, and a loss of transparency. Such modifications facilitate the contamination of removable orthodontic appliances and aligners with microorganisms [20]. Thus, the current literature lacks data that clearly show which approach is ideal for cleaning clear aligners. Based on their personal experiences, orthodontists and dental hygienists recommend a variety of techniques, including the use of toothbrushes, toothpaste, tablets, and even certain odd techniques, such as the use of vinegar, descaling solutions, or washing liquid [21].

### 4.1. Surface Morphology and Biofilm Accumulation

The SEM analysis in our study revealed that Invisalign aligners demonstrated a superior structure with a smoother and less defective surface than that using three systems of clear aligners [22]. Moreover, a SEM characterization study of two generations of clear aligners conducted in 2021 demonstrated that the polymeric materials used in the fabrication of clear aligners exhibited chemical stability even after being subjected to accelerated aging in the laboratory. The intraoral use of aligners was the primary factor contributing to most of the reported modifications [23]. Thus, it is essential that any orthodontic appliance inserted into the oral cavity in clinical practice should be free of bacteria and clean. A recent SEM study by Low et al. [24] revealed the susceptibility of invisible aligners to colonization. They found that more recessed and sheltered areas of the appliance, such as the cusp tips and attachment dimples, had higher biofilm accumulation and featured biofilms with complex structures.

Trials involving acrylic resin removable devices were performed [21,25]. One of the earliest investigations examined the microbial populations found in removable orthodontic appliances after tooth brushing, self-acting washing, and ultrasound treatment [25]. SEM analysis was used to assess bacterial colonization following the application of these three techniques. The results show that a toothbrush and toothpaste alone did not decontaminate the device, whereas ultrasound treatment eliminated plaque accumulation in particular regions. However, none of the tested methods completely decontaminated detachable appliances [25], and the cleaning crystal solution procedure had no effect on the inhibition of bacterial colonization in another study. The highest level of bacterial biofilm adhesion was noted after routine brushing of the aligners. When a vibrating bath was combined with a crystal cleaning solution, the maximum reduction in biofilm adherence and accumulation on the posterior aligner (50%) was achieved [11].

In the present study, tooth brushing alone or in combination with a whitening toothpaste resulted in noticeable debris retention on the surfaces of the tested aligners. In contrast, the aligners cleaned with vinegar, Fittydent Super Cleansing Tablets, and Invisalign cleaning crystals exhibited significantly lower debris accumulation. These findings suggest that chemical-based cleaning methods may exhibit superior efficacy in terms of debris removal than mechanical cleaning approaches, potentially providing a more thorough surface cleansing of clear aligners.

The Invisalign cleaning crystal solution used in this study had no detrimental effects on the aligner during the test period. According to the manufacturer, the Invisalign cleaning crystal solution contains strong cleaning agents, such as sodium sulfate (which helps break down deposits), sodium carbonate (cleans and removes stains), sodium tripolyphosphate (breaks down organic material), sodium dichloroisocyanurate (exerts antimicrobial action by releasing chlorine), and surfactants (loosens debris and exerts stain removal action). Although these components did not have any adverse effects on the aligners during the test period, they may cause some detrimental effects after prolonged usage, especially sodium dichloroisocyanurate, which releases chlorine and may cause slight degradation [26]. Notably, some patients reported an unpleasant taste of chlorine when using Invisalign cleaning crystals, which may affect user preference. In addition, the SEM images revealed that the polyurethane’s surface was harmed by ultrasonic vibrations, displaying obvious evidence of ultrasonic cavitation and water absorption [27]. Such damage may have serious repercussions, including an increase in the likelihood of bacterial adhesion owing to surface imperfections [28].

Levrini et al. [10,21] demonstrated that the use of toothpaste alone was more effective than washing with tap water, whereas the use of toothpaste in conjunction with an effervescent tablet yielded better outcomes. Thus, effervescent tablets improve cleaning effectiveness when used in conjunction with toothpaste. Nevertheless, biofilms were not entirely eliminated, regardless of the addition of effervescent tablets. Consequently, the brushing method appears to be crucial for regulating biofilm formation. Compared with previous studies reporting partial effectiveness of brushing, effervescent tablets, and ultrasound treatment [10,21,25], the present findings support the superior cleaning performance of chemical-based methods, particularly, Invisalign cleaning crystals and whitening toothpaste, in maintaining aligner surface cleanliness without compromising material integrity.

Although our SEM evaluation focused on debris accumulation, bacterial biofilms were not directly assessed in this study. This is a limitation of the current work, as microbial colonization plays a significant role in aligner hygiene. Prior studies have shown that peroxide-based solutions, like those in Invisalign cleaning crystals and Fittydent tablets, can reduce bacterial biofilms when combined with mechanical agitation [11,25], though complete decontamination is rarely achieved. Future research involving microbiological sampling is needed to confirm the antibacterial effectiveness of these cleaning agents.

### 4.2. Color Stability Using Spectrophotometry and NBS Ratings

Studies have examined the elimination of bacterial biofilms from aligners and retainers using different cleaning solutions and procedures [27,29]; however, few have compared the transparency of the devices after cleaning [4]. A previous study found that effervescent tablets containing sodium carbonate and sulfate, when combined with brushing, represented the best method for cleaning orthodontic aligners, although biofilms remained on the internal surfaces, which could cause discoloration, odor, and synergistic interactions [21]. The observed color-change variations are attributed to the surface characteristics of the materials [30]. These changes are attributed to characteristics, such as roughness, which may enhance pigment accumulation. This may explain the color modification caused by both the penetration of the material and its effect on the surface [31]. Research has indicated that the translucency of thermoplastic plates diminishes following thermoforming, which is attributed to the transition from amorphous to crystalline structures influenced by the temperature, pressure, and processing duration. These investigations indicated a negligible color change following thermoforming [32].

The polar nature of polyurethane, in conjunction with the surface porosity of the aligners, was hypothesized to be the cause of their susceptibility to staining [4,33]. The penetration of pigments from the external environment into the polymer is facilitated by increased water absorption [7,34]. Aligners that had been exposed to tea for seven days nearly recovered their original color after cleaning with Invisalign cleaning crystals, thus suggesting that the technique is effective for removing stains. The Invisalign appliances examined, which were stained as a result of exposure to coffee or red wine for 12 h or 7 days, continued to exhibit a significant color change following cleaning with Invisalign cleaning crystals and Retainer Brite Cleaning Tablets containing sodium carbonate and sodium sulfate. Researchers have confirmed that cleansers have superior stain removal potential for tea compared with other chromogenic agents, such as those found in coffee and red wine [4]. The present findings revealed that Invisalign cleaning crystals and Fittydent Super Cleansing Tablets demonstrated comparable efficacy in terms of stain removal. However, brushing with the whitening toothpaste exhibited superior performance in effective stain removal. This was also confirmed by the NBS ratings, where four cleaning methods—vinegar, Fittydent Super Cleansing Tablets, Invisalign cleaning crystals, and water—led to noticeable color changes in the aligners after cleaning. In contrast, brushing with whitening toothpaste resulted in a perceivable color change, indicating a distinct difference in its ability to maintain the clarity of aligners compared to other methods.

### 4.3. Mechanical Properties (Nanoindentation)

Polyurethane possesses a variety of interesting characteristics, including high elasticity, flexibility, chemical resistance, oxidation resistance, mechanical strength, and ease of processing [7]. The thermoplastic polyurethane used in the Invisalign aligners exhibited a high degree of hardness and high elastic modulus in a previous study on the mechanical and chemical properties of aligners [35]. The findings of this study align with those of a prior study reporting no significant differences between the tested cleaner groups and the control group for Invisalign aligners, as indicated by the loading and unloading curves [3]. Invisalign aligners are resistant to the degradative effects of cleaning agents [3], thus indicating that the evaluated cleaning methods did not adversely affect the elastic modulus of the material. These results suggest that cleaning processes maintain the structural integrity of the aligner material, supporting its safe use in clinical settings.

### 4.4. Chemical Stability

An FTIR analysis was performed to describe the chemical alterations induced on the aligner surfaces. The FTIR-ATR analysis of the samples treated with the five cleaning methods revealed consistent spectral patterns across all groups. Peak characteristics of secondary amines (N–H stretching at 3310 cm^−1^), N–H bending (1600 cm^−1^), C=O stretching (1698 cm^−1^), CH_2_ (1411 cm^−1^), and C–O stretching (1216 cm^−1^) were consistently observed in all the analyzed samples. These findings reinforce the idea that cleaning protocols do not compromise the chemical composition of the aligner materials. These findings are consistent with those of previous studies, highlighting the chemical stability of Invisalign aligners subjected to various cleaning methods, which support their continued use in clinical practice [3]. Moreover, the present findings agree with those of recent studies that identified a polyurethane-based material in the tested clear aligner samples using the Invisalign system [7,22,35].

### 4.5. Clinical Implications and Recommendations

The findings indicate that Invisalign cleaning crystals can be safely used for routine aligner cleaning without jeopardizing the material’s integrity. Furthermore, whitening toothpaste may be considered a better option for people concerned about aligner discoloration. As no major mechanical or chemical changes were found, these cleaning procedures can be employed with confidence without compromising aligner function or durability, hence enhancing patient compliance and treatment success.

### 4.6. Limitations and Future Research Directions

Although this study offers valuable data concerning the short-term effects of cleaning agents, certain limitations must be considered. For instance, this study was performed in vivo and did not include structured dietary assessments and variations in patients’ dietary habits, such as the consumption of colored or acidic beverages, which may have influenced aligner degradation. Furthermore, the sample size was limited to 60 samples from 12 patients, which may not adequately represent variability within a larger population. Furthermore, this study evaluated a limited number of commercially available cleaning agents, indicating that other potential cleaning methods should be investigated to provide a more comprehensive understanding of their effects on aligner integrity. In addition, elemental surface analysis and microbiological assessments were not conducted. Identifying specific bacterial species would require immediate post-use collection and processing under controlled laboratory conditions, which was not feasible in this clinical setting. Future research with a broader range of cleaning agents is warranted to better assess the impacts of various cleaning protocols on Invisalign aligners.

## 5. Conclusions

This study evaluated the effects of five different cleaning methods on Invisalign clear aligners in an in vivo setting using SEM, spectrophotometry, FTIR, and nanoindentation analyses. The results demonstrate that Invisalign cleaning crystals had no detrimental effects on the aligner surface during the study period. Spectrophotometric analysis using NBS ratings indicated noticeable color changes in the aligners cleaned with vinegar, Fittydent Super Cleansing Tablets, Invisalign cleaning crystals, and water, whereas brushing with whitening toothpaste resulted in a perceptible color change. The elastic modulus remained unchanged across all the tested groups, suggesting that none of the cleaning methods affected the mechanical integrity of the aligners. The FTIR analysis confirmed the absence of significant chemical alterations in the aligner material. These findings provide valuable insight into the efficacy and safety of different cleaning methods for clear aligners.

## Figures and Tables

**Figure 1 polymers-17-01620-f001:**
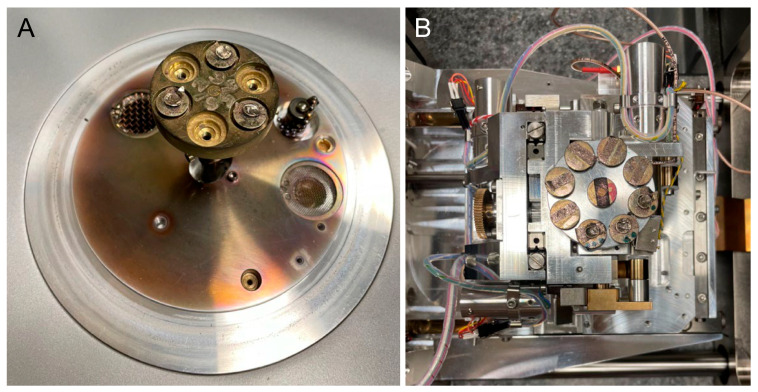
(**A**) Samples were gold-coated with a quorum sputter coater for scanning electron microscopy imaging. (**B**) Samples were loaded for surface morphology examination using a Zeiss device (EVO-LS10, Cambridge, UK).

**Figure 2 polymers-17-01620-f002:**
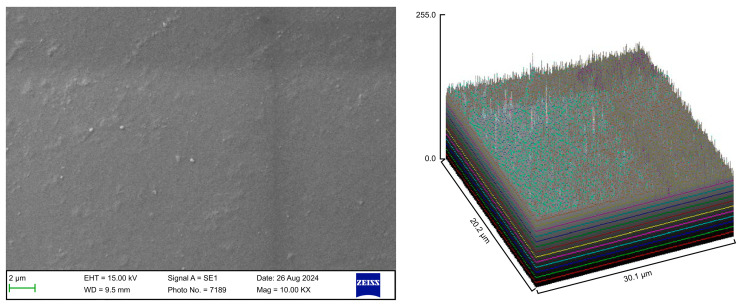
Scanning electron microscopy images of the control group.

**Figure 3 polymers-17-01620-f003:**
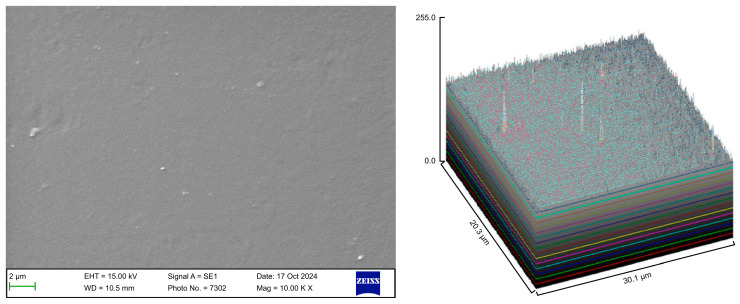
Scanning electron microscopy images of Group A.

**Figure 4 polymers-17-01620-f004:**
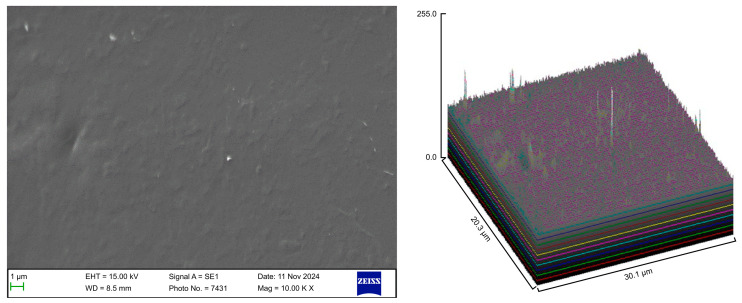
Scanning electron microscopy images of Group B.

**Figure 5 polymers-17-01620-f005:**
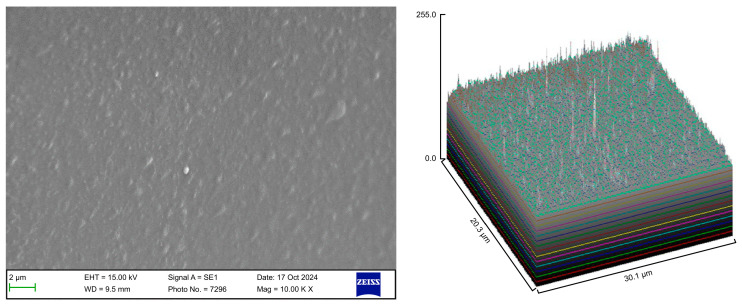
Scanning electron microscopy images of Group C.

**Figure 6 polymers-17-01620-f006:**
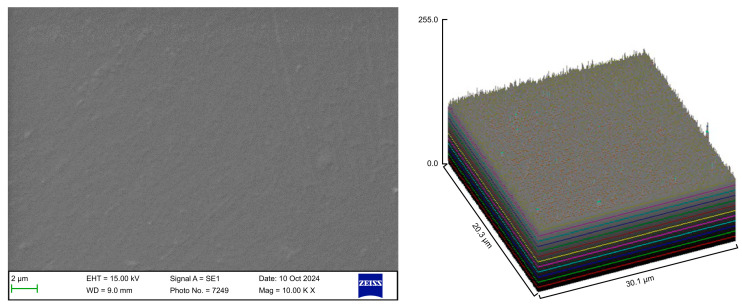
Scanning electron microscopy images of Group D.

**Figure 7 polymers-17-01620-f007:**
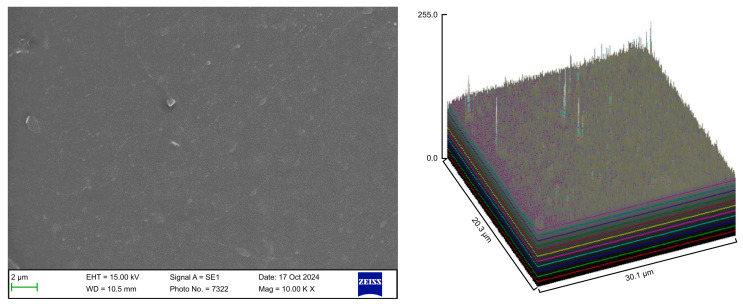
Scanning electron microscopy images of Group E.

**Figure 8 polymers-17-01620-f008:**
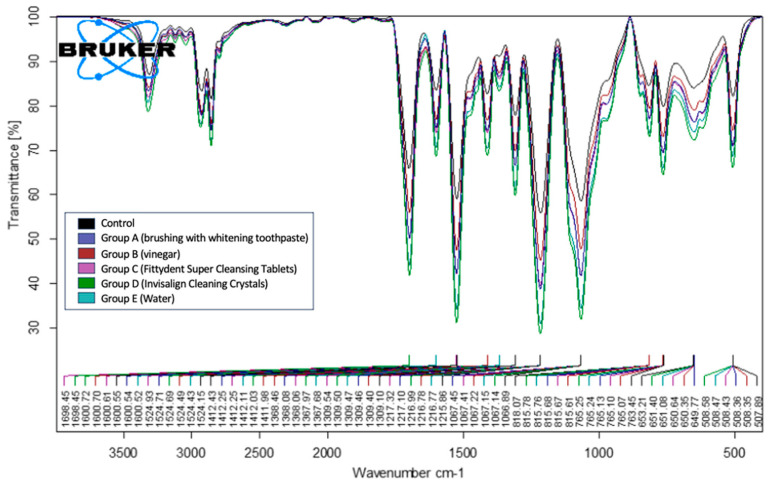
Fourier transform infrared spectroscopy-attenuated total reflectance data of the five study groups showing all the peaks that are similar to the standard peak.

**Table 1 polymers-17-01620-t001:** NBS ratings.

NBS Units	Descriptions of the Color Changes
0.0–0.5	Trace: Extremely slight change
0.5–1.5	Slight: Slight change
1.5–3.0	Noticeable: Perceivable
3.0–6.0	Appreciable: Marked change
6.0–12.0	Much: Extremely marked change
12.0 or more	Very much: Change to another color

NBS: National Bureau of Standards.

**Table 2 polymers-17-01620-t002:** The surface roughness parameters (in nanometers) of the aligner samples from the different groups.

Parameters	Control	Group A	Group B	Group C	Group D	Group E
Rq	52.5455	3.3432	48.1625	48.5843	55.9103	54.9019
Ra	25.3871	2.3377	21.7543	22.7522	27.361	25.9476
Rt	257.1719	86.7526	257.5824	259.465	257.6126	256.8808

Rq: RMS roughness, Ra: average roughness, Rt: maximum profile height.

**Table 3 polymers-17-01620-t003:** Comparison of the mean ΔE values among the study groups.

Study Groups	ΔE		
	Mean (Sd.)	F-Value	*p*-Value
Group A (brushing with whitening toothpaste)	3.21 (1.91)	8.661	<0.0001
Group B (vinegar)	7.15 (4.87)		
Group C (Fittydent Super Cleansing Tablets)	4.86 (3.48)		
Group D (Invisalign cleaning crystals)	5.17 (3.98)		
Group E (water)	5.61 (4.05)		

ΔE: color changes.

**Table 4 polymers-17-01620-t004:** Color changes converted to NBS units and remarks on the color differences.

Study Groups	NBS	
	Mean (Sd.)	Color Change
Group A (brushing with whitening toothpaste)	2.42 (1.87)	Noticeable
Group B (vinegar)	6.35 (4.79)	Appreciable
Group C (Fittydent Super Cleansing Tablets)	4.06 (3.45)	Appreciable
Group D (Invisalign cleaning crystals)	4.38 (3.95)	Appreciable
Group E (water)	4.79 (3.98)	Appreciable

NBS: National Bureau of Standards.

**Table 5 polymers-17-01620-t005:** Comparison of the mean elastic modulus values between each study group and the control group.

Study Groups	Elastic Modulus		Mean Difference	t-Value	*p*-Value
	Mean (Sd.)	Control Group Mean (Sd.)			
Group A (brushing with whitening toothpaste)	0.8357 (0.17)	0.8546 (0.02)	−0.0188	−0.369	0.716
Group B (vinegar)	0.8938 (0.48)	0.8546 (0.02)	0.0392	0.273	0.788
Group C (Fittydent Super Cleansing Tablets)	0.8673 (0.28)	0.8546 (0.02)	0.0127	0.151	0.882
Group D (Invisalign cleaning crystals)	0.9508 (0.38)	0.8546 (0.02)	0.0962	0.831	0.416
Group E (water)	0.7646 (0.18)	0.8546 (0.02)	−0.0899	−1.660	0.113

## Data Availability

The original contributions presented in this study are included in the article. Further inquiries can be directed to the corresponding authors.

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
