# Peer review of "The Effectiveness of Different Cleaning Methods for Clear Orthodontic Aligners: Impacts on Physical, Mechanical, and Chemical Properties—An In Vivo Study"

_polymers, 2025, doi:10.3390/polym17121620_

Round 1
Reviewer 1 Report
Comments and Suggestions for Authors
In the paper, the authors report effectiveness of different cleaning methods for clear orthodontic aligners: impact on physical, mechanical, and chemical properties—an in vivo study. SEM confirmed the effectiveness of Invisalign Cleaning Crystals in maintaining cleanliness, revealing a surface similar to that of the control group with no adverse effects. Color stability analysis revealed significant ΔE value differences; whitening toothpaste had significantly lower ΔE values than water and Invisalign Cleaning Crystals. The elastic modulus and FTIR analyses indicated no significant differences between the cleaning methods. The paper is well written. However, there are some problems needed to be analyzed.
- The component of whitening toothpaste, vinegar, Fittydent Super Cleansing Tablets and Invisalign Cleaning Crystals should be further measured or discussed.
- The surface property of orthodontic aligners should be further measured such as surface elements and bacterial kinds.
- The performance of clearance in this paper should be compared with other reported methods.
- The effect of whitening toothpaste, vinegar, Fittydent Super Cleansing Tablets and Invisalign Cleaning Crystals on the bacterial biofilms should be further measured or discussed.
Reviewer 2 Report
Comments and Suggestions for Authors
This study evaluated the effects of five different cleaning methods on Invisalign clear aligners in an in vivo setting using SEM, spectrophotometry, FTIR, and nanoindentation analyses. The results show that Invisalign Cleaning Crystals and whitening toothpaste are safe for aligner maintenance, providing valuable insight into the efficacy and safety of different cleaning methods for clear aligners. I have no objection to this manuscript.
Reviewer 3 Report
Comments and Suggestions for Authors
This study evaluated the effects of five different cleaning methods on Invisalign clear aligners in an in vivo setting using SEM, spectrophotometry, ATR-FTIR, and nanoindentation analyses. This contribution could bring more clinically related information about cleaning clear aligners. Several issues should be resolved before further consideration.
- Line 129. The authors should provide the rationale for choosing 10 days instead of the 14 days reported in the literature (Line 382). Further, what is the frequency of cleaning the aligner in each group? Are they all the same, three times per day, as they eat the meals?
- Table 2. The authors should identify the unit of the three roughness parameters. Are they in the micron order?
- Besides the statement made from Lines 294-305, the authors are advised to use the post hoc analyses following the ANOVA analysis shown in Table 3. This would make it clearer to the readers the relative statistical differences among these five groups and the control.
- Figure 8. The authors should label the color used for the different treatment groups.
- Section 4.4. The authors should comment on the implications of the reduced absorption intensity of various ATR-FTIR peaks noted in Figure 8.
- Line 518. The authors have stated that one of the limitations of this study is the variations in dietary habits. For these twelve patients, it would be easier for the authors to identify whether there are variations in dietary habits in their clinical visits/follow-ups. The authors must report more information on these dietary habits in this contribution.
Round 2
Reviewer 1 Report
Comments and Suggestions for Authors
The authors have addressed the problem very well, and the manuscript can be accepted in the present form.
Reviewer 3 Report
Comments and Suggestions for Authors
The authors have addressed the issues raised in the earlier submission.